# One-year continuation of postpartum intrauterine contraceptive device: Findings from a prospective cohort study in India

**Ashish Srivastava[1‡], Surendra Sharma[2‡], Kamlesh Lalchandani[1], Nochiketa Mohanty[3], Deepak Chandra Bhatt[1]***, Gulnoza Usmanova[1], Bulbul Sood[1], Somesh Kumar[1]**

**1** Jhpiego-An Affiliate of Johns Hopkins University, New Delhi, India, **2** Jhpiego-An Affiliate of Johns Hopkins University, Raipur, Chhattisgarh, India, **3** Jhpiego-An Affiliate of Johns Hopkins University, Bhubaneshwar, Odisha, India

‡ AS and SS are co-first authors.
* deepakbhatt001@gmail.com

## Abstract

**Data Availability Statement:** All relevant data are within the manuscript and its Supporting Information files.

### Objective(s)

To estimate continuation rates for postpartum intrauterine contraceptive device (PPIUD) at 6 weeks, 6 months and 1-year within existing programs in an under-resourced setting, and to identify determinants of discontinuation, removal and expulsion.

### Study design

We used a prospective cohort design and enrolled recent PPIUD adopter women across 100 public healthcare facilities in Odisha and Chhattisgarh, India. We collected their socio-demographic information and followed them up telephonically at 6 weeks, 6 months and 1 year for complications and continuation status. We assessed PPIUD continuation rates and factors associated with PPIUD discontinuation, removal, and expulsion using Cox proportional hazards modelling.

### Results

We enrolled 916 participants (579 (63.2%) from Odisha and 337 (36.8%) from Chhattisgarh). The continuation rate of PPIUD was 88.7% at 6 weeks, 74.8% at 6 months 60.1% at one year. Once discontinued, chances of not opting for any family planning method was high (up to 81.2%). Participants with education of 6th to 12th class and those experiencing complications (pain abdomen, bleeding and discharge per vaginum) were more likely to remove the IUD with adjusted hazard ratio of 1.82 (95% CI: 1.18–2.79) and 4.39 (95% CI: 3.25–5.93) respectively. For expulsion, we did not find any factor that was statistically significant.

**Funding:** This work is supported by a philanthropic non-profit foundation committed to improvement of population health. It prefers to remain unnamed. The funders had no role in study design, data collection and analysis, decision to publish, or preparation of the manuscript.

**Competing interests:** The authors have declared that no competing interests exist.

## Conclusion(s)

PPIUD continuation rates declined considerably after the initial 6 weeks. Counselling and follow-up services for managing complications must be strengthened, especially in the first 6 weeks of PPIUD insertion, to enhance and sustain programmatic impact.

## Implications

Our findings emphasize on the need to strengthen client counseling and follow-up for management of complications, especially in the first 6 weeks of insertion of PPIUDs. Ongoing programs need to address comprehensive capacity building efforts in this regard.

## 1. Introduction

The Sustainable Development Goals (SDG 3.7) have committed to universalize access to reproductive health and family planning services by 2030 [1]. The 'Every Woman Every Child' Global Strategy (2016–2030) has further prioritized unwanted pregnancy as a health challenge necessitating evidence-based research interventions [2].

Modern contraceptives help in improving reproductive health and outcomes [3,4]. Intrauterine contraceptive devices (IUDs) are among the most effective and commonly used modern contraceptives available [5,6]. Offering IUDs to women immediately postpartum (immediately after or within 48 hours of delivery; postpartum IUD (PPIUD)) has advantages as institutional delivery—(a) offers a window of opportunity for IUD insertion; (b) women are more likely to accept an IUD immediately postpartum; and (c) several women may not revisit the health facility once discharged, but become sexually active soon after [7].

Considering these global evidences and potential advantages, the Government of India introduced PPIUD services within its family planning program in 2010–2011 [8]. The services were introduced at district level and above facilities (District Hospitals and Medical Colleges), and subsequently scaled up to sub-district level facilities like Sub-District Hospitals, Community Health Centers and select Primary Health Centers. This scale up has resulted in a multifold increase in the number of acceptors of PPIUDs, from about 0.2 million in FY 2013–14 to about 2.5 million in FY 2019–20 [9].

The Family Planning Quality of Care framework–also known as the Bruce/Jain framework —is a widely accepted framework for assessment of the quality of family planning services in Low- and Middle-income countries. It identifies six elements of family planning programs that together constitute quality: choice of methods, information given to users, technical competence, inter- personal relations, follow-up or continuity mechanisms, and appropriate constellation of services. It further links these elements to outcomes like improved client knowledge, satisfaction, health and contraceptive use. Under contraceptive use, it considers both acceptance and continuation of FP method as key outcomes of quality service provision [10]. Therefore, in context of the PPIUD program in India, which has seen a notable increase in acceptance since inception, retention or continuation of PPIUD to provide desired spacing is equally important to understand.

While there are a few studies from India which evaluated the continuation rate of PPIUDs, they were either conducted in highly controlled environments or at well-resourced tertiary care centers [11–13]. As government scales up the services to primary and secondary level

health facilities, it is imperative to know and analyze continuation rates, and determinants thereof, for PPIUDs at sub-tertiary level facilities in real life programmatic settings.

We undertook the study in two Empowered Action Group (EAG) states in India–Chhattisgarh and Odisha. EAG states are those having poor reproductive, maternal, newborn, child and adolescent health (RMNCAH) indicators and high unmet needs of family planning [14].

The primary objective of the study was to estimate continuation rates for PPIUD over a period of one-year (at 6 weeks, 6 months and 1 year) within existing programs operational at sub-district level facilities Secondary objectives included identifying and quantifying the reasons cited by clients for discontinuation of PPIUDs and for identifying the determinants (client profile e.g. socio-demographic and reproductive factors, chief complaints, and the service provisioning factors) of PPIUD continuation, for potential programmatic prioritization through the public health system in similar resource constrained settings.

## 2. Materials and methods

### Study setting

The Jhpiego-led 'Expanding Access to Intrauterine Contraceptive Device Services in India' (EAISI) project (Phase I: 2014–17) strengthened the delivery of family planning (FP) services in Chhattisgarh and Odisha (two EAG states) in partnership with State Governments and the Ministry of Health and Family Welfare (MoHFW), GoI.

Odisha is a state on the eastern coast while Chhattisgarh is a state in central India. At the time of the study (2015–16), Odisha and Chhattisgarh had an estimated population of 43.1 million and 27.5 million, with 17.6% and 24.9% population residing in urban areas, respectively [15]. The total fertility rate was 2.1 and 2.2 while the modern contraceptive prevalence rates were 45.4% and 54.5%, respectively [16]. In terms of literacy rate and poverty, in both states, the female literacy levels were below the national average (68.4%) while the proportion of population multi-dimensionally poor were much higher than the national average (24.85%) (**Table 1**) [16,17].

### Intervention package

EAISI aimed at increasing contraceptive choice for WRAs at 187 select public-sector sub-district health facilities (Odisha: n-118; Chhattisgarh: n-69) by expanding access to LARC through establishment of postpartum IUCD, interval IUCD, post-abortion IUCD and family planning (FP) counseling services, and by training the providers therein.

Under EAISI, all eligible women received family planning counseling during antenatal, intra-natal and post-partum periods by health service providers (Staff Nurses, Doctors, FP counsellors). Individualized family planning counseling was administered via one-on-one consultations. PPIUD was offered to all willing women delivering in the public health facilities after obtaining consent and due confirmation of eligibility i.e., expressed desire to limit their

**Table 1. State profiles in comparison to the national profile.**

| Indicator | Chhattisgarh | Odisha | India |
|---|---|---|---|
| Population (Census) | 27,571,000 | 43,128,000 | 12,90,235,000 |
| Proportion of urban population to the total estimated population (2016) | 24.9% | 17.6% | 32.8% |
| Literacy rate (Female) (NFHS4) | 66.3% | 67.4% | 68.4% |
| Proportion of population which is multidimensionally poor (NITI aayog (2023)) | 29.90% | 29.34% | 24.85% |
| TFR (NFHS 4) | 2.2 | 2.1 | 2.2 |
| MCPR (NFHS4) | 54.5% | 45.4% | 47.8% |

fertility or space the next pregnancy by at least one year, history of regular menstruation, no present history or visible signs of reproductive tract infections, and no history of postpartum hemorrhage during the current delivery. Trained healthcare workers (doctors and nurses) inserted the PPIUDs. The insertion was done using the Kelly's forceps after normal vaginal deliveries while it was done manually, before closing the endometrium in case of caesarean sections. All woman received a Copper T 380A. The program maintained a detailed database of PPIUD clients. The other post-partum family planning options available through the national family planning program during the time of the study were post-partum sterilization and condoms.

## Study design, sampling and sample size

We used a prospective observational study design to identify a cohort of 1000 WRAs who had received PPIUD at one of the sampled sites during or within a month before October 2015. For representativeness, we used a two-stage cluster sampling technique. For the first stage of sampling, in October 2015, we enlisted all EAISI project supported health facilities in Odisha and Chhattisgarh that had at least five insertions of PPIUD in the preceding month. From these we selected 100 sites (63 from the state of Odisha and 37 from Chhattisgarh) through simple random sampling while retaining the proportional representation of the states (1.7:1). For the second stage, from each facility, trained facility staff enrolled 10 consecutive WRAs. The WRAs were identified from among those who had accepted PPIUD in September/ October 2015, consented to participate in this prospective cohort and be followed-up telephonically for up to one-year post insertion of the PPIUD, and could provide a valid mobile number for telephonic follow-up. The sample size (1000 participants across 100 sites) was majorly chosen for ease of operationalization; it allowed absolute power (power = 100%) to estimate extremely high continuation rates (above 99% assuming alpha = 0.05 after initial insertion). One of our earlier retrospective studies on PPIUD continuation suggested that PPIUD continuation rates were about 60% at one-year [18]. Thus, even if there would have been about 50% loss-to-follow-up at the end of Year 1, the sample size would have been good enough to assess PPIUD continuation rates of around 60% with an absolute precision of 6%, design effect of 2.0 and at 95% confidence level (n = [DEFF*Np(1-p)]/ [(d2/Z21-α/2*(N-1)+p*(1-p)]).

## Data collection tools

While the participants were identified from among the beneficiaries of the ongoing EAISI program, we designed the cohort with its own metrics for measurement of outcomes. For this, a semi-structured bilingual (English and local language; Odia for Odisha and Hindi for Chhattisgarh) questionnaire was used for data collection by telephone, at baseline and in subsequent follow-ups. The questionnaire was designed after a review of relevant literature, pretesting on 10 PPIUD acceptors each in Odisha and in Chhattisgarh at facilities which were not included in this study, and finalized in consensus among the authors by editing/ replacing phrases that were difficult to administer and/ or comprehend. The questionnaire included 14 questions (sociodemographic characteristics: education, income and number of family members; experience with PPIUD: continued use, removal, expulsion, time interval between insertion and removal or expulsion, associated side effects, reason for insertion, reason for removal, switch to other FP methods, and reasons for discontinuation). Socio-economic status was calculated using BG Prasad's scale [19]. Data on age of the clients, number of living children, type of IUD insertion and provider's cadre were extracted from the EAISI program database. IUD insertion which was performed within 10 minutes of expulsion of placenta was regarded as post-

placental insertion while those performed after 10 minutes but within 48 hours of delivery were regarded as immediate post-partum insertion.

## Data collection procedure

Please refer to S1 Appendix.

## Statistical analysis

We managed data on Microsoft Excel spreadsheets and analyzed using SPSS software, version 24.0 [20]. We represented descriptive statistics as frequency and proportions (categorical data) or central tendencies and dispersion (continuous data). For knowing the statistical significance of difference in distribution of categorical data, we used the chi-square test.

We used Kaplan-Meier time-to-event probabilities to estimate the cumulative continuation rates at every 45-day interval. It utilizes all follow-up time contributed by the participants, including time to discontinuation (due to expulsion or removal) as well as incomplete follow-up time as censored event time. We used self-reported time of removal or expulsion, together with date of insertion, to calculate the time to event in days. We considered self-reported removals and expulsions as discontinuation events, regardless of whether the woman then took up a similar or different contraceptive method. We censored participants who had not reported discontinuation at their last completed follow up–for participants who were categorized as lost to follow up, censoring time was kept as the time interval between date of insertion and date of last follow up; for participants who reported continuing with the method at one year follow up, the censoring time was kept as 365 days.

To estimate the risk of PPIUD discontinuation, we undertook Cox proportional hazards modeling for identifying predictors of removal and expulsion, initially unadjusted and then by adjusting variables pragmatically into the model. Cox proportional hazards models are recommended in analysis of prospective studies which have a follow up period during which occurrence of events is observed. This is because they have more statistical power than other models as they account for time until events occur [21,22].

For describing the socio-demographic characteristics of the study participants, estimating continuation probabilities (Kaplan-Meier analysis) and estimating the risk of PPIUCD discontinuation (Cox proportional hazards modelling), we utilized the complete study data set. For conducting further analysis on removals and expulsions, we utilized sub-sets of the study data set, which included data of study participants who had reported the respective events, within one year of insertion.

Statistical significance was identified at p<0.05 for the analyses.

## Ethics

The study protocol was approved by the International Institute of Health Management Research (IIHMR), New Delhi, India (letter no. IIHMR-D/IRB. NO. 3/2015-16 dated October 13, 2015), Research & Ethics Committee, Directorate of Health Services, Odisha (letter no. 227/SHRMU dated October 5, 2015 and the Institutional Review Board of Johns Hopkins Bloomberg School of Public Health (Application no 6537 dated October 19, 2015)). Respective state governments granted administrative approval for the study. Our efforts were in compliance with the Indian Council of Medical Research Ethical Guidelines of 2017 [23].

## Results

We enrolled 916 PPIUD recipients from our intervention sites who consented to participate in telephonic follow ups for next one year. Of the 916 participants, 579 (63.2%) were from Odisha and 337 (36.8%) from Chhattisgarh; the participation ratio for the two states aligned with their sampling proportions (i.e. 1.7:1) (**Table 2**).

**Table 2. Distribution of participants according to socio-demographic characteristics, provider type and associated side-effects experienced ($N$ = 916).**

| Characteristic | Number (%) |
|---|---|
| **State** | |
| Odisha | 579 (63.2) |
| Chhattisgarh | 337 (36.8) |
| **Age of the respondent** | |
| < = 25 years | 620 (67.7) |
| > 25 years | 237 (25.9) |
| Data Missing | 59 (6.4) |
| **Education of the respondent** | |
| Illiterate/ just literate | 208 (22.7) |
| Up to 5th standard | 91 (9.9) |
| 6- 12th standard | 566 (61.8) |
| Graduate/Post-graduate | 46 (5.1) |
| Data Missing | 05 (0.5) |
| **Number of living children*** | |
| < 2 | 452 (49.3) |
| > = 2 | 405 (44.2) |
| Data Missing | 59 (06.4) |
| **Socio economic status** | |
| Lower class | 179 (19.5) |
| Lower middle class | 364 (39.7) |
| Middle class | 206 (22.5) |
| Upper/ upper middle class | 91 (10.0) |
| Missing | 76 (08.3) |
| **Reason for accepting PPIUD** | |
| Spacing | 537 (58.6) |
| Limiting | 362 (39.5) |
| Don't know | 17 (01.9) |
| **Type of service provider who inserted the PPIUD** | |
| Doctor | 193 (21.1) |
| Nurse | 664 (72.5) |
| Data Missing | 59 (06.4) |
| **Type of insertion** | |
| Intra-caesarean insertion | 15 (01.6) |
| Post placental insertion | 239 (26.1) |
| Post-partum insertion | 603 (65.8) |
| Data Missing | 59 (06.4) |

*Family planning initiatives in India motivate beneficiaries to limit family size to two children.

Abbreviation: PPIUD, post-partum intrauterine device.

**Table 3. Cumulative continuation rates of PPIUD at 45-day time intervals among the participants.**

| Time interval (days) | Number of participants | | | Survival % (95% CI)* |
|---|---|---|---|---|
| | Entering interval | Reporting terminal event (removal / expulsions) | Lost to follow up (cumulative) (% of total lost to follow up) | |
| 0–45 (6 weeks) | 916 | 102 | 15 | 88.7 (86.5–90.6) |
| 45–90 | 799 | 43 | 40 | 83.8 (81.3–86.1) |
| 90–135 | 716 | 52 | 18 | 77.7 (74.8–80.3) |
| 135–180 (6 months) | 646 | 24 | 10 (83) (43.9%) | 74.8 (71.7–77.5) |
| 180–225 | 612 | 34 | 65 | 70.4 (67.2–73.3) |
| 225–270 | 513 | 23 | 20 | 67.2 (63.8–70.2) |
| 270–315 | 470 | 25 | 13 | 63.5 (60.1–66.8) |
| 315–365 (1 year) | 432 | 21 | 08 (189) (100%) | 60.1 (56.5–63.5) |

By end of 1-year follow-up, 189 (20.6%) participants were lost-to-follow-up (including 83 (09.1%) at 6 months). Participants that were lost-to-follow-up had similar (p > 0.05) location, age and socio-economic status profile as that of other study participants. However, there was a significant difference in their education status, with higher proportion of lost to follow up participants being lesser educated. (S1 Table).

Overall, 403 participants reported continuing with their method at one year. The cumulative continuation probability of PPIUD was 88.7% (95% CI: 86.5–90.6) at 6 weeks, 74.8% (95% CI:71.7–77.5) at 6 months and 60.1% (95% CI: 56.5–63.5) at one year (Table 3).

Of the 916 enrolled PPIUD acceptors, 110 reported expulsions while 214 reported removals, within one year of insertion. Majority expulsions (58.1%) were reported within 6 weeks of insertion, while majority removals (41.6%) happened in the 6 months to 1-year interval (S2 Table).

We collected information about reasons for removal from the 214 participants who reported the same. Almost half (48.1%) of those who got the PPIUD removed reported that they did so due to symptoms like bleeding, pain in abdomen or discharge; another 18.7% underwent removal to switch to another method of contraception (Table 4).

Once discontinued, chances of not opting for any FP method was high; 67.3% among those who got it removed and 81.2% in those in whom the device got expulsed (Table 5).

**Table 4. Primary reason cited for removal of PPIUD by participants who got their PPIUD removed during the follow-up period i.e., within one year of insertion.**

| Reason for removal | N (%) |
|---|---|
| Had symptoms like bleeding per vaginum/ pain in abdomen/ vaginal discharge | 103 (48.1) |
| Husband/ family was against it | 10 (04.7) |
| Wanted to have another child | 07 (03.3) |
| Wanted to use some other contraceptive method | 40 (18.7) |
| PPIUD got displaced/partially expelled | 11 (05.1) |
| Fear & misconception related to PPIUD | 09 (04.2) |
| Failure | 01 (0.4) |
| Other reasons | 33 (15.4) |
| **Total** | **214 (100.0)** |

Abbreviation: PPIUD, post-partum intrauterine device.

**Table 5. Contraceptive method taken up by the participants whose PPIUDs had to be removed or got expulsed during the follow-up period i.e., within one year of insertion.**

| Contraceptive methods | Post removal (N = 214) n (%) | Post expulsion (N = 110) n (%) | Total (N = 324) n (%) |
|---|---|---|---|
| Condoms | 15 (7.0) | 06 (5.5) | 21 (6.5) |
| Oral contraceptive pills | 19 (8.9) | 08 (7.3) | 27 (8.3) |
| Male permanent contraception | 01 (0.5) | 0 | 01 (0.3) |
| Female permanent contraception | 34 (15.9) | 05 (4.5) | 39 (12.0) |
| Intrauterine device | 01 (0.5) | 01 (0.9) | 02 (0.6) |
| Did not use any modern contraceptive | 144 (67.3) | 90 (81.8) | 234 (72.2) |

Abbreviation: PPIUD, post-partum intrauterine device.

Participants with education of 6[th] to 12[th] class and those experiencing complications (pain abdomen, bleeding and discharge per vaginum) were more likely to remove the PPIUD with adjusted hazard ratio of 1.82 (95% CI: 1.18–2.79) and 4.39 (95% CI: 3.25–5.93), respectively. For expulsion, we did not find any factor that was statistically significant (**Table 6**).

## Discussion

We report that continuation rate of PPIUD was 88.7% at 6 weeks, 74.8% at 6 months and 60.1% at 1 year. It dropped maximally within the first 6 weeks of insertion. Among those who got the PPIUD removed, almost half reported associated side effects like bleeding, pain in abdomen and vaginal discharge, to be the primary reason. On conducting Cox proportional hazards modeling, woman's education status and experiencing associated side-effects emerged as significant determinants of PPIUD removal; we could not find any significant determinant for PPIUD expulsion.

Our estimates of PPIUD continuation rates either align or report an improvement over that in recent literature; [12,13,24–28] and aligns with that reported by our team from retrospective data in an earlier study [18]. However, the study also does highlight that a majority of participants who got the device removed, reported expected side effects like discharge and pain in abdomen as the primary reason for removal. This corroborates with findings from other studies conducted in India [18,29,30] and emphasizes the need to include discussion on these expected side effects during the counseling. There are studies from India and other low- and middle-income countries, which have shown gaps in the quality of family planning counseling, specifically with regards to discussion on expected side effects of methods [31,32].

It is also concerning to report that a large proportion of women did not opt for any other modern contraceptive following discontinuation, suggesting possible poor client satisfaction and grossly unmet needs. At the time of the study, only three options- Condoms, PPIUDs or post-partum sterilization were available for post-partum FP at the selected public health facilities.

The lack of switching may also point towards the fact that those who had removals because of the side effects (majority of removals were due to experiencing side effects), may have preferred a gap before switching to another method. It must be acknowledged that the assessment of switching among discontinuers was limited by the fact that they were not followed up any further once they had reported discontinuation–hence women who would have taken some time to switch after discontinuation, may not have been captured.

Since, average discontinuation rates were maximal within 6 weeks of insertion and the women were likely to continue without a contraception thereafter, care and follow-up within

**Table 6. Factors associated with PPIUD removal and expulsion among the participants.**

| Factor | Hazards Ratio for Removal (95% CI); N = 214 | | Hazards Ratio for Expulsion (95% CI); N = 110 | |
|---|---|---|---|---|
| | Unadjusted | Adjusted* | Unadjusted | Adjusted* |
| **Age of the client** | | | | |
| < = 25 years | 1.0 | 1.00 | 1.00 | 1.00 |
| >25 years | 1.1 (0.8–1.6) | 1.0 (0.7–1.4) | 0.70 (0.43–1.12) | 0.64 (0.37–1.11) |
| **Education of the client** | | | | |
| Illiterate/ Just literate | 1.0 | 1.0 | - | - |
| Up to 5th standard | 1.3 (0.7–2.4) | 1.3 (0.7–2.5) | - | - |
| 6th - 12th standard | **1.8 (1.3–2.7)** | **1.8 (1.2–2.8)** | - | - |
| Diploma/ Graduate/ Post-graduate | 1.2 (0.6–2.4) | 1.42 (0.6–3.4) | - | - |
| **Reason for accepting PPIUD** | | | | |
| Limiting | 1.0 | 1.0 | - | - |
| Spacing | 0.9 (0.7–1.2) | 0.9 (0.6–1.3) | - | - |
| **Status of Family Planning counselling before accepting method** | | | | |
| Not counselled/ counselled only on PPIUD | 1.0 | 1.0 | - | - |
| Counselled on one or more other methods | 1.0 (0.6–1.6) | 1.1 (0.5–2.1) | - | - |
| **Number of living children** | - | - | 0.98 (0.78–1.22) | 1.08 (0.83–1.39) |
| **Type of insertion** | | | | |
| Immediate post-partum | - | - | 1.00 | 1.00 |
| Post-placental | - | - | 0.96 (0.62–1.47) | 0.97 (0.63–1.49) |
| **Type of service provider who inserted PPIUD** | | | | |
| Doctor | 1.0 | 1.0 | 1.00 | 1.00 |
| Nurse | 1.2 (0.8–1.7) | 1.4 (1.0–2.1) | 0.96 (0.61–1.52) | 0.90 (0.56–1.42) |
| **Socio-economic status** | | | | |
| Upper/upper middle class | 1.0 | 1.0 | - | - |
| Middle class | 0.8 (0.5–1.3) | 0.8 (0.5–1.4) | - | - |
| Lower middle class | 0.8 (0.5–1.3) | 0.9 (0.6–1.5) | - | - |
| Lower class | 0.8 (0.5–1.3) | 1.1 (0.6–2.0) | - | - |
| **Experienced pain in abdomen/ bleeding/ discharge during use** | | | | |
| No | 1.0 | 1.00 | - | - |
| Yes | **3.6 (2.8–4.7)*** | **4.4 (3.5–5.9)** | - | - |

*Cox proportional hazards model adjusted for variables in the table for which the hazards ratios have been mentioned.

Statistically significant (p<0.05) values have been emboldened.

Abbreviation: PPIUD, post-partum intrauterine device.

the first 6 weeks postpartum ('window of opportunity') need strengthening. Robust follow-up and improved capacity for managing associated side-effects (includes effective counseling skills) may help in salvaging high removal rates due to complications.

It is encouraging to note that PPIUD expulsion rates were not significantly impacted by provider profile (e.g., doctor, nurse). This finding corroborates with previous studies that have found no significant difference between expulsion rates in insertions done by doctors and nurses, or significantly fewer expulsions when insertions were done by nurses versus the doctors [27,33]. This gives confidence in FP task shifting and sharing to providers and facilities further downstream, which can further improve access to PPIUD services. A significant proportion of institutional deliveries in India are conducted by nurses and midwives (including auxiliary nurse midwives and general nurse midwives) [18,34] and therefore they can play a key role in enabling access to PPIUD services, in light of these findings.

The study also found no significant differences in expulsion rates by type of insertion, a finding which is corroborated by earlier studies [27,35]. It is worth noting that we could not include manual placement following caesarean section as a category for the 'type of insertion' variable, as there were very few (15/916) such cases and none of those insertions had an expulsion, as reported till the last follow up.

The non-linear role of education in PPIUD continuation calls for further exploration. Education shapes health and care seeking related attitudes, intentions and behaviors [36,37]. However, female education and its manifestations in LMIC settings is an outcome of complex inter-linkages between socio-cultural milieu and other structural determinants of overall societal development and gender empowerment. PPIUD uptake and continuation among women in LMIC contexts have been shown to require multi-layered and multi-dimensional interventions [38]. Many studies from India have highlighted the key role of partners and family members in the decision to take up and use a family planning method [30,39,40]. Our study also found that some participants opted for removal primarily because of their partner's/husband's disapproval. This emphasizes the need for increased engagement with family and partners during counseling.

Estimates from 2019 show that reproductive and sexual health services grossly lag behind needs in LMICs–about 1.6 billion WRAs live in these countries, 885 millions of whom want to avoid pregnancy but of which about 218 million (~24.6%) are not using any modern contraception–annually, about 111 million pregnancies (49%) in LMICs are unwanted [41]. Latest (2020) modelled estimates suggest that despite attempts at rapid expansion in the FP services, these will not match the even faster increasing unmet needs [42]. Hence, while services must be scaled up fast, the progresses made thus far must sustain its coverage and quality and not 'backslide' [43,44]. Our study underscores the importance of strengthening client counseling, follow-up and management of associated side-effects, especially in the first six weeks of insertion of PPIUDs. There are opportunities within existing programmatic outreach services (e.g., immunization sessions) in India and in other similar LMIC contexts that can be leveraged through effective task sharing and shifting with workers in the outreach. Focusing on quality of FP services through comprehensive capacity building efforts will help in enhancing client satisfaction and sustaining confidence in modern methods of contraception. We hope that program managers working in other resource constrained settings will relate to the study's context and use the evidence generated by it to strengthen their respective PPIUD services and FP programs.

## Strengths and limitations

The main strengths of our study are its prospective design that collected data within a large-scale public health program in mid and primary level care facilities of two high priority states in India. Thus, the results of our study represent real-life challenges unlike in a highly monitored (controlled) research study setting. A notable strength of our study lies in its unique setting. It was conducted at sub-district level peripheral health facilities as opposed to high-capacity tertiary healthcare centers–which serve as study sites for majority of published studies from India. Therefore, our study contributes significantly to addressing an existing evidence gap, specifically regarding the decentralization of family planning services to downstream health facilities.

The health facility staff did the recruitment for the study; this minimized refusal rates (the participant trusted the provider).

Our study had the following limitations: (1) we could not recruit all PPIUD acceptors in the EAISI program–only those who had consented and provided a telephone number were

recruited to enable follow-up; (2) All recruited acceptors could not be followed-up as some telephone numbers were invalid, some could not be reached and some did not respond during subsequent follow-ups (20.6% clients were lost-to-follow-up by the 3rd follow-up). On comparing the background characteristics of those who were lost to follow up versus those who were successfully followed up, we did find a significant difference in their education status. Higher proportion of women who were lost to follow up during this study, had lesser education when compared with women who could be successfully followed up. This may have had an influence on the estimation of the PPIUD continuation probability, as education status did emerge as a significant risk for PPIUD removals in our study; (3) We could not explore the reasons for not choosing another contraception among participants who discontinued PPIUD and did not switch to another method. We operated within public health systems that were supported by a matured and evidence-informed project (EAISI) leading to potentially improved PPIUD continuation rates. Further qualitative studies can help in developing a better understanding on the issue; (4) Telephonic follow-ups implicated data validation challenges and we had to rely on self-reported outcomes.

## Supporting information

**S1 Table. Comparison of socio-demographic characteristics between participants successfully followed up and those lost to follow-up.**
(DOCX)

**S2 Table. Reported number of expulsions and removals by time intervals.**
(DOCX)

**S1 Appendix.**
(DOCX)

## Acknowledgments

We are most grateful and extend our sincere thanks to all study participants, governments of Chhattisgarh and Odisha, our donor and colleagues, and Dr Archisman Mohapatra.

## Author Contributions

**Conceptualization:** Ashish Srivastava, Surendra Sharma, Kamlesh Lalchandani, Deepak Chandra Bhatt, Bulbul Sood, Somesh Kumar.

**Data curation:** Ashish Srivastava, Deepak Chandra Bhatt.

**Formal analysis:** Ashish Srivastava, Deepak Chandra Bhatt, Gulnoza Usmanova.

**Funding acquisition:** Kamlesh Lalchandani, Bulbul Sood, Somesh Kumar.

**Investigation:** Ashish Srivastava, Surendra Sharma.

**Methodology:** Ashish Srivastava, Deepak Chandra Bhatt, Gulnoza Usmanova.

**Project administration:** Surendra Sharma, Kamlesh Lalchandani, Nochiketa Mohanty, Bulbul Sood, Somesh Kumar.

**Resources:** Kamlesh Lalchandani, Bulbul Sood, Somesh Kumar.

**Software:** Ashish Srivastava.

**Supervision:** Ashish Srivastava, Kamlesh Lalchandani, Nochiketa Mohanty, Deepak Chandra Bhatt, Somesh Kumar.

**Validation:** Ashish Srivastava, Deepak Chandra Bhatt, Gulnoza Usmanova.

**Visualization:** Deepak Chandra Bhatt.

**Writing – original draft:** Ashish Srivastava, Surendra Sharma, Kamlesh Lalchandani, Nochiketa Mohanty, Deepak Chandra Bhatt.

**Writing – review & editing:** Ashish Srivastava, Surendra Sharma, Kamlesh Lalchandani, Nochiketa Mohanty, Deepak Chandra Bhatt, Gulnoza Usmanova, Bulbul Sood, Somesh Kumar.

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
