## [Decision Letter · Decision Letter 0]

16 Apr 2023

PONE-D-22-32704One-year continuation of postpartum intrauterine contraceptive device: findings from a prospective cohort study in IndiaPLOS ONE

Dear Dr. BHATT,

Thank you for submitting your manuscript to PLOS ONE. After careful consideration, we feel that it has merit but does not fully meet PLOS ONE’s publication criteria as it currently stands. Therefore, we invite you to submit a revised version of the manuscript that addresses the points raised during the review process.

We look forward to receiving your revised manuscript.

Kind regards,

Rajesh Raushan, PhD

Academic Editor

PLOS ONE

3. Please provide additional details regarding participant consent. In the ethics statement in the Methods and online submission information, please ensure that you have specified whether: 1) whether the ethics committee approved the verbal/oral consent procedure, 2) why written consent could not be obtained, and 3) how verbal/oral consent was recorded. If your study included minors, please state whether you obtained consent from parents or guardians in these cases. If the need for consent was waived by the ethics committee, please include this information.

4. We note that the grant information you provided in the ‘Funding Information’ and ‘Financial Disclosure’ sections do not match. When you resubmit, please ensure that you provide the correct grant numbers for the awards you received for your study in the ‘Funding Information’ section.

“This work is supported by a philanthropic non-profit foundation committed to improvement of population health. It prefers to remain unnamed.”

“This work is supported by a philanthropic non-profit foundation committed to improvement of population health. It prefers to remain unnamed. The funders had no role in study design, data collection and analysis, decision to publish, or preparation of the manuscript.”

**Comments to the Author**

1. Is the manuscript technically sound, and do the data support the conclusions?

Reviewer #1: Yes

Reviewer #2: Yes

2. Has the statistical analysis been performed appropriately and rigorously? 

Reviewer #1: Yes

Reviewer #2: I Don't Know

3. Have the authors made all data underlying the findings in their manuscript fully available?

Reviewer #1: Yes

Reviewer #2: Yes

4. Is the manuscript presented in an intelligible fashion and written in standard English?

Reviewer #1: Yes

Reviewer #2: No

5. Review Comments to the Author

Reviewer #1: Paper is well sequenced. Title is justifying the paper appropriately. Methods are chosen appropriately. Results and discussions are crisp and sound. The paper is recommendable. paper can be accepted as it is.

Reviewer #2: Thank you for your article which is interesting and shows important information regarding continuation rates of PPIUD in the two states in India. There are some parts of the paper which need the English revising. I have not done this as part of my review.

These are my comments regarding the content:

Line 71-74 Correct English

In the introduction I suggest you write a paragraph on the two states involved in your study. You need to explain their social leconomic and geographical factors ie. are they predominanty rural or urban, are women travelling long distances to access health, what is the terrain like, are their issues like floods / mountain ranges etc… that would make access to health difficult? Are they similar states in terms of this information or do they differ. Maybe include a table to compare standard paramenters?

Method

There is no comment on the methodology used to insert the PPIUD. Please explain further – was it using the Kelly’s forceps, or manually, wrigleys or purpose built inserter. This has an impact on expulsion rates and so needs to be explained. Also no mention of which IUDs were being inserted. Were they copper, if so which exact type.

There is no mention of the counselling service provided initially to the women – was it during ANC or intra partum or post partum. Who did the counselling – designated counsellors, nurses, doctors? Was the counselling individualised or group counselling. This has a big impact on continuation rates and so needs to be explained clearly.

There is no mention of the contraceptive options that women had in the post partum period. Was it PPIUD or nothing? Or PPIUD or sterilisation? Or were other methods LARCs avaulbale like implants? Or short acting eg POP. Please explain as this has a big influence on reason for choice of method and likely reasons for discontinuation.

Discussion

Overall I would say the discussion portion of your paper is weak. I think if you expand more on the context and the counselling aspects of the project you may be able to discuss your results abit more meaningfully.

Line 190 – English

Line 188 – 192 the paragraph does not discuss anything - it’s just a repetition of the results. Please either add some discussion to this or remove.

Line 192. There is a lack of discussion regarding the expulsion rates and whether they may vary according to method of insertion. What was the method? Did the method vary between units? Please look at: https://obgyn.onlinelibrary.wiley.com/doi/full/10.1002/ijgo.12600 One of the countries is India. Expulsion rates are different according to type of insertion. If you don’t have the data to look at this ten you must mention this as a limitation in the discussion.

IN paragraph starting line 194, you say the discontinuation rates are similar to what has been reported before. Perhaps this is true in India but necessarily true elsewhere. Please see this paper from Tanzania: https://reproductive-health-journal.biomedcentral.com/articles/10.1186/s12978-020-00999-4

I would argue your discontinuation rates are quite high and it worrying that such a large proportion of those who discontinued do not take up any modern contraception. I think you need to discuss this further. Is this because there is no other reliable method available free of charge? Or is this because they wanted to have more children and so don’t want any contraception, or is this because the side effects were so awful that they have been put off using any method of contraception. You don’t mention consent for PPIUCD. Was this taken properly? Or could the IUCD have been inserted without proper consent. Although you mention complex issues I think you need to go in depth a little more. I would add a paragraph explaining the context better – is contraception widely accepted in these areas of India? Or are there cultural barriers, do husbands approve generally, are women empowered to make decisions about contraception or do their husbands/mother-in-laws make that decision. All these aspects need to be mentioned and discussed. If you don’t have the data or cant find anything in the literature then you have to put this in your discussion as a weakness and suggest further qualitative studies need to be undertaken to understand exactly why the discontinuation rates are so high.

This paper looks at counselling aspects and acceptance in the first place of PPIUD: https://obgyn.onlinelibrary.wiley.com/doi/full/10.1002/ijgo.12599

Line 225

The loss to follow up proportion at 1 year follow up is quite high (60%). Please make loss to follow up at different stages more explicit, perhaps in a table. You also need to look at the demographics of those lost to follow up to see if they are different to the overall group so that you can estimate if there may be a degree of bias in your analysis due to this. Please include this.

Line 229 – 231

What do you mean? Did you ask all those who discontinued to come to a health center to have face to face follow up? If so, you need to add this in your methodology and talk about the results of these follow ups. If you don’t have the results of these follow up appointments then you cannot say they validate your findings.

Table 1

Type of insertion – please explain your categories: post placental and postpartum.

Table 2

How can your continuation rate be 60.4% overall if your expulsion and removal rates are 13.7% and 29.9% respectively (totalling 43.6% which would give you a continuation rate of 56.4??). The maths doesn’t add up. Please explain.

Table 5

Why didn’t you look at expulsion following intra caesarean insertion. There are studies showing expulsion rates are lower following intra-cesarean insertion compared to within 10 mins of a normal vaginal delivery. Please explain why not or add analysis.

6. PLOS authors have the option to publish the peer review history of their article (what does this mean?). If published, this will include your full peer review and any attached files.

Reviewer #1: No

Reviewer #2: No

---

## [Author Response · Author response to Decision Letter 0]

15 Sep 2023

Reviewer 1- Paper is well sequenced. Title is justifying the paper appropriately. Methods are chosen appropriately. Results and discussions are crisp and sound. The paper is recommendable. paper can be accepted as it is.

Response: Thank you so much for your comments and appreciation.

Reviewer 2: Thank you for your article which is interesting and shows important information regarding continuation rates of PPIUD in the two states in India. There are some parts of the paper which need the English revising. I have not done this as part of my review.

Response: Thank you for your valuable inputs and comments. They have helped improve the manuscript a lot. We have attempted to incorporate them to the best of our capacity and hopefully the revised version meets your expectation

Query: Line 71-74 Correct English

In the introduction I suggest you write a paragraph on the two states involved in your study. You need to explain their social economic and geographical factors i:e. are they predominantly rural or urban, are women travelling long distances to access health, what is the terrain like, are their issues like floods / mountain ranges etc.… that would make access to health difficult? Are they similar states in terms of this information or do they differ. Maybe include a table to compare standard parameters?

Response: As suggested, we have now included a paragraph about the states and their comparison to national parameters, in order to give readers a context of the study sites. 

Method

There is no comment on the methodology used to insert the PPIUD. Please explain further – was it using the Kelly’s forceps, or manually, wriggles or purpose built inserter. This has an impact on expulsion rates and so needs to be explained. Also no mention of which IUDs were being inserted. Were they copper, if so which exact type.

There is no mention of the counselling service provided initially to the women – was it during ANC or intra partum or post partum. Who did the counselling – designated counsellors, nurses, doctors? Was the counselling individualised or group counselling. This has a big impact on continuation rates and so needs to be explained clearly.

There is no mention of the contraceptive options that women had in the post-partum period. Was it PPIUD or nothing? Or PPIUD or sterilisation? Or were other methods LARCs available like implants? Or short acting eg POP. Please explain as this has a big influence on reason for choice of method and likely reasons for discontinuation.

Response: As suggested, we have now included a paragraph about the states and their comparison to national parameters, in order to give readers a context of the study sites. 

Thank you for this feedback. We have now included all these important details in the material and methods section. 

Discussion

Overall I would say the discussion portion of your paper is weak. I think if you expand more on the context and the counselling aspects of the project you may be able to discuss your results abit more meaningfully.

Line 190 – English

Line 188 – 192 the paragraph does not discuss anything - it’s just a repetition of the results. Please either add some discussion to this or remove.

Line 192. There is a lack of discussion regarding the expulsion rates and whether they may vary according to method of insertion. What was the method? Did the method vary between units? Please look at: https://obgyn.onlinelibrary.wiley.com/doi/full/10.1002/ijgo.12600 One of the countries is India. Expulsion rates are different according to type of insertion. If you don’t have the data to look at this ten you must mention this as a limitation in the discussion.

Response: Thank you very much for your valuable inputs on the discussion section. We have attempted to incorporate them.

Line 190: We have reframed the sentence.

Line 188-192 – In the first paragraph of the discussion we have summarized the key results with reference to our study objectives. This is in line with the STROBE checklist/guidelines for reporting of observational studies. 

As has now been detailed out, mostly all insertions were done using modified Kelly’s forceps, following a vaginal delivery. Very few insertions were done manually after caesarean sections. Only 15 of the 916 cases were of manual insertions after a caesarean section, and none of them reported an expulsion. Therefore, we were not able to include intra caesarean category for ‘type of insertion’ variable in the regression model for expulsion – we have now included this as part of the discussion. 

In addition, as has been shown in table 5, the expulsion rates did not vary by cadre of provider or by timing of insertion. We have now added some discussion points around this finding.

Query: IN paragraph starting line 194, you say the discontinuation rates are similar to what has been reported before. Perhaps this is true in India but necessarily true elsewhere. Please see this paper from Tanzania: https://reproductive-health-journal.biomedcentral.com/articles/10.1186/s12978-020-00999-4

I would argue your discontinuation rates are quite high and it worrying that such a large proportion of those who discontinued do not take up any modern contraception. I think you need to discuss this further. Is this because there is no other reliable method available free of charge? Or is this because they wanted to have more children and so don’t want any contraception, or is this because the side effects were so awful that they have been put off using any method of contraception. You don’t mention consent for PPIUCD. Was this taken properly? Or could the IUCD have been inserted without proper consent. Although you mention complex issues I think you need to go in depth a little more. I would add a paragraph explaining the context better – is contraception widely accepted in these areas of India? Or are there cultural barriers, do husbands approve generally, are women empowered to make decisions about contraception or do their husbands/mothers-in-law make that decision. All these aspects need to be mentioned and discussed. If you don’t have the data or cant find anything in the literature then you have to put this in your discussion as a weakness and suggest further qualitative studies need to be undertaken to understand exactly why the discontinuation rates are so high.

This paper looks at counselling aspects and acceptance in the first place of PPIUD: https://obgyn.onlinelibrary.wiley.com/doi/full/10.1002/ijgo.12599

Response: We compared the expulsion rates reported at the time of study. Thanks for Tanzania paper, its interesting and captures interesting finding. Please note that while as per law, women are the sole decision makers as far as contraception goes in India, social factors as mentioned by you definitely play a role. We have now included this in our discussion. As part of the program, before insertion of PPIUDs, consent was taken by the provider – we have added this detail in the methods section. 

Method switching was not a focus of our study and definitely need more research and we agree with your observation – we have added a suggestion on conducting qualitative studies to delve deeper into this subject.

Although the continuation rates are comparable to other studies from India, analyzing the reasons for removals does point towards a possible gap pertaining to the quality of counseling on the method. Majority of those who got the device removed reported expected side effects like vaginal discharge and pain in abdomen to be the primary reasons for opting for removal. This aligns with findings of earlier studies which too reported expected side effects to be the primary reason for removal. This emphasizes the need to include discussion on these transient and expected side effects during the counseling on the method and prepare women for it. This could help further improve continuation rates of the method. It is also important to highlight that around 5% clients reported pressure from their partners as the primary reason for discontinuing the method. Some recent studies from India have also pointed out family pressure to be an important reason for removal of PPIUCDs. This emphasizes the need to involve partners and even family members like mother in laws in the discussions around family planning during different stages of the pregnancy. 

The lack of switching among those who discontinued the method (either because of expulsion or removal), could also be because of limited options available for post-partum contraception in India’s FP program – at the time of the study only two other options- either condoms or PPS – were available for post-partum FP. It may also point towards the fact that those who experienced the side effects (majority of removals were due to experiencing side effects), may not have immediately wanted to switch to another method. It should be acknowledged that the status of switching among discontinuers was limited by the fact that they were not followed up further once they had reported discontinuation of the method in any of their scheduled follow ups – hence women who would have taken some time to switch may not have been captured. We have now included all these aspects in the discussion section, citing relevant literature.

Line 225

The loss to follow up proportion at 1 year follow up is quite high (60%). Please make loss to follow up at different stages more explicit, perhaps in a table. You also need to look at the demographics of those lost to follow up to see if they are different to the overall group so that you can estimate if there may be a degree of bias in your analysis due to this. Please include this.

Line 229 – 231

What do you mean? Did you ask all those who discontinued to come to a health centre to have face to face follow up? If so, you need to add this in your methodology and talk about the results of these follow ups. If you don’t have the results of these follow up appointments then you cannot say they validate your findings.

We would like to clarify that the loss to follow up at one year was 20.6%. This has been corrected in the discussion section (where we are including this as a limitation) and has also been stated in the initial part of the results section. We would also like to add, that as has been mentioned in the sample size estimation paragraph, even if our study had seen a 50% lost to follow up rate, the sample size would have been sufficient. 

As suggested, we have now included numbers on loss to follow up at each stage, in the newly added table no. 2. 

Distribution of clients with respect to follow up status in the study – 

1. Outcome known at one year – 727 (79.4%)

2. Lost to follow up prior to 6 months (at 2nd follow-up) – 83 (9.06%)

3. Lost to follow up after 6 months (at 3rd follow-up) – 106 (11.57%)

We have compared the demographics of those lost to follow up versus those who participated throughout (Supplemental table 1) and found no significant differences by location, age and SE status. However, there was a significant difference with respect to education, with higher proportion of respondents who were lost to follow up being lesser educated. This may have influenced the study estimates of continuation rates (discontinuation rates could be higher) as the study did find education status as a significant determinant of removals. We have now included this as a limitation in the discussion section. 

Line 229-231 – We agree. We have clarified the context now. All those who had reported discontinuation during their follow ups were encouraged to follow up with health service providers at the health facility, post completion of data collection interview. These follow ups were not a part of the study and as suggested, we have reframed the sentence to ensure it does not reflect a data validation exercise. 

Table 1

Type of insertion – please explain your categories: post placental and postpartum.

Postplacental insertion refers to insertion performed within 10 min of expulsion of placenta

following a vaginal delivery whereas postpartum insertion is done after 10 min but within 48 h of delivery.

We have added this to the manuscript.

Table 2

How can your continuation rate be 60.4% overall if your expulsion and removal rates are 13.7% and 29.9% respectively (totalling 43.6% which would give you a continuation rate of 56.4??). The maths doesn’t add up. Please explain.

As we had used the life table analysis method, these are probabilities of each event (continuation, removal or expulsion) happening at specified time intervals along with their confidence intervals. They may therefore not add up to 100%. 

In order to avoid any confusion regarding this, we have now kept only the life table analysis for continuation (newly added table 2)in the main manuscript. This also depicts lost to follow up status at different intervals. We have included details on number of removals and expulsions in the narrative and also added a supplemental table 2 pertaining to it. 

Table 5

Why didn’t you look at expulsion following intra caesarean insertion. There are studies showing expulsion rates are lower following intra-caesarean insertion compared to within 10 mins of a normal vaginal delivery. Please explain why not or add analysis.

As has been previously explained, the sample includes very few PPIUCDs that were inserted after a caesarean section and therefore we are unable to comment on expulsion rates following caesarean sections. The study was conducted in sub-district level facilities and as women requiring caesarean sections are usually referred to district or higher level facilities (very few sub-district level facilities conduct caesarean deliveries), the sample consists of very few post caesarean section PPIUCDs. 

We have added this to the discussion section.

---

## [Decision Letter · Decision Letter 1]

23 Oct 2023

PONE-D-22-32704R1One-year continuation of postpartum intrauterine contraceptive device: findings from a prospective cohort study in IndiaPLOS ONE

Dear Dr. Bhatt,

Thank you for submitting your manuscript to PLOS ONE. After careful consideration, we feel that it has merit but does not fully meet PLOS ONE’s publication criteria as it currently stands. Therefore, we invite you to submit a revised version of the manuscript that addresses the points raised during the review process.

ACADEMIC EDITOR:The manuscript seems improved. But there is scope for the improvement before moving the acceptance for the publication. Some of those needed are-

1. Introduction section needs expansion with background, context/need of the study and framed objectives accordingly.

2. Data and Method section needs improvements. The table number 1 and their interpretation should be move to result part as an starting section.

3. Although there details about the sample size and design including sample framework used in the selection of samples but it should be very clearly mentioned under the sub-heading of 'sampling design and sample size'.

4. The paragraphs in this section are large needs to divide into smaller paragraph with lined and readable flow.

5. The data collection procedure can be moved to the appendix part of the manuscript.

6. The statistical methods mentioned in line number 176-184 needs to revise and strengthen. As on the modelling part authors mentioned about using Cox modelling needs appropriate scientific exploration and valid justification for the use of the such hazard modeling procedure for this study.

7. Ethical Clarence should be in under separate heading.

8. In the result section the description of the tables seems very loosely written needs proper interpretation.

9. Table number also needs correction as table no 1 is in method section as well as in the result section. In the table 1 of result section, change the word client to respondents. What methods the authors have followed to classify the respondents into various socio-economic status category.

10. Heading of table number 2- 'life table'- what does it mean for this specific table. The word should be changed with the appropriate words which should reflect the actual meaning.

11. The sample size varies in different tables which should be clearly mentioned in sample design section of under the methodology part of the manuscript.

12. The discussion section also needs restructuring with clear cut findings of the study followed by the suggestive measures as the study seems programme based outcome linked to FP and unmet needs which remained an important concern before the government and programme strategic concern. That would be worth of the study and knowledge addition of the study.

13. Limitations of study is although mentioned in discussion section but needs proper paragraph.

14. Careful editing is needed of the whole manuscript.

Thank You!!! Be sure to:Indicate which changes you require for acceptance versus which changes you recommendAddress any conflicts between the reviews so that it's clear which advice the authors should followProvide specific feedback from your evaluation of the manuscriptPlease ensure that your decision is justified on PLOS ONE’s publication criteria and not, for example, on novelty or perceived impact.

We look forward to receiving your revised manuscript.

Kind regards,

Rajesh Raushan, PhD

Academic Editor

PLOS ONE

Journal Requirements:

Reviewers' comments:

Reviewer's Responses to Questions

**Comments to the Author**

1. If the authors have adequately addressed your comments raised in a previous round of review and you feel that this manuscript is now acceptable for publication, you may indicate that here to bypass the “Comments to the Author” section, enter your conflict of interest statement in the “Confidential to Editor” section, and submit your "Accept" recommendation.

Reviewer #2: All comments have been addressed

2. Is the manuscript technically sound, and do the data support the conclusions?

Reviewer #2: Yes

3. Has the statistical analysis been performed appropriately and rigorously? 

Reviewer #2: I Don't Know

4. Have the authors made all data underlying the findings in their manuscript fully available?

Reviewer #2: Yes

5. Is the manuscript presented in an intelligible fashion and written in standard English?

Reviewer #2: Yes

6. Review Comments to the Author

Reviewer #2: THANK YOU FOR TAKING INOT ACCOUNT MY POINTS – YOUR CHANGES ARE APPROPRIATE AND HAVE MUCH IMPROVED THE PAPER. IT NOW READS VERY WELL.

7. PLOS authors have the option to publish the peer review history of their article (what does this mean?). If published, this will include your full peer review and any attached files.

Reviewer #2: **Yes: **ANITA MAKINS

---

## [Author Response · Author response to Decision Letter 1]

3 Apr 2024

1. Introduction section needs expansion with background, context/need of the study and framed objectives accordingly. Response:- We have expanded the background section, clearly bringing out the context and the need for this study. We have specified the primary and secondary objectives of the study towards the end of the background section. 

2.Data and Method section needs improvements. The table number 1 and their interpretation should be move to result part as an starting section. Response:- As was suggested by one of the reviewers, we had introduced this table in the methods section to describe the study settings in more detail. This is not a finding from the study but a detailed description of the state profile and its comparison to the national profile – hence we feel it is more appropriate in the methods section describing the study setting. We have corrected the table numbers in the results section as suggested. 

3.Although there details about the sample size and design including sample framework used in the selection of samples but it should be very clearly mentioned under the sub-heading of 'sampling design and sample size'. Response:-As suggested, we have now divided the methods section into different sub-sections with Study design, sampling and sample size being one of them. 

4.The paragraphs in this section are large needs to divide into smaller paragraph with lined and readable flow. Response:- As suggested we have now divided the methods section into different sub-sections. We have also further divided the larger paragraphs into smaller paragraphs for improved readability.

5.The data collection procedure can be moved to the appendix part of the manuscript. Response:-As suggested, we have moved the data collection procedure to an appendix (S1 Appendix). 

6.The statistical methods mentioned in line number 176-184 needs to revise and strengthen. As on the modelling part authors mentioned about using Cox modelling needs appropriate scientific exploration and valid justification for the use of the such hazard modeling procedure for this study. Response:-As suggested, we have now strengthened the statistical analysis section detailing out the process of estimating cumulative continuation probabilities as well as adding a justification for using Cox proportional hazards modelling. The justification added is as following – ‘Cox proportional hazards models are recommended in analysis of prospective studies which have a follow up period during which occurrence of events is observed. This is because they have more statistical power than other models as they account for time until events occur. (line 212-215)

7.Ethical Clarence should be in under separate heading. Response:-As suggested, it is now under a sub section on Ethics. (Line 224- 230) 

8. In the result section the description of the tables seems very loosely written needs proper interpretation. Response:-We have done interpretation of the results in the Discussion section, which is as per various guidelines on writing different sections of a scientific manuscript. In the results section, we have narrated the key findings from each table. 

9.Table number also needs correction as table no 1 is in method section as well as in the result section. In the table 1 of result section, change the word client to respondents. What methods the authors have followed to classify the respondents into various socio-economic status category. Response:- We have corrected the table numbers. 

As suggested, we have changed the term to respondents. 

The BG Prasad scale was used for classifying respondents into various socio-economic status categories. The same has been mentioned in the methods section (Line 187 in clean version)

10.Heading of table number 2- 'life table'- what does it mean for this specific table. The word should be changed with the appropriate words which should reflect the actual meaning. Response:-We used Kaplan-Meier time-to-event probabilities to estimate the cumulative continuation rates. It utilizes all follow-up time contributed by the participants, including time to discontinuation (due to expulsion or removal) as well as incomplete follow-up time as censored event time. We used self-reported time of removal or expulsion, together with date of insertion, to calculate the time to event in days. We considered self-reported removals and expulsions as discontinuation events, regardless of whether the woman then took up a similar or different contraceptive method. We censored participants who had not reported discontinuation at their last completed follow up – for participants who were categorized as lost to follow up, censoring time was kept as the time interval between date of insertion and date of last follow up; for participants who reported continuing with the method at one year follow up, the censoring time was kept as 365 days. 

As suggested, we have now replaced the term ‘Life table analysis’ with ‘Cumulative continuation rates.’ We have added the above details in the Statistical Analysis section.

11.The sample size varies in different tables which should be clearly mentioned in sample design section of under the methodology part of the manuscript. Response:-The sample size varies for tables which further analyse expulsions and removals – like reasons for removal and switching after either expulsion or removal. For such tables, the sample consists of study participants who reported these events from among all participants. We have included this information in the narrative of results section. 

We have now also added a paragraph in the statistical analysis section regarding the same.

12.The discussion section also needs restructuring with clear cut findings of the study followed by the suggestive measures as the study seems programme based outcome linked to FP and unmet needs which remained an important concern before the government and programme strategic concern. That would be worth of the study and knowledge addition of the study. Response:- As suggested, we have re-structured the discussion section to bring the suggestive measures after discussion on key findings of the study. 

We have separated out the paragraph which discussed the strengths and limitations of the study, and kept it at the end. 

13.Limitations of study is although mentioned in discussion section but needs proper paragraph. Response:- As suggested, we have now kept a separate paragraph on ‘Strengths and Limitations’ in the latter half of the Discussion section.

14.Careful editing is needed of the whole manuscript. Response:-We have re-read and made some additional copy-edits. Hope this meets the expectation.

---

## [Editor Report · Decision Letter 2]

7 May 2024

One-year continuation of postpartum intrauterine contraceptive device: findings from a prospective cohort study in India

PONE-D-22-32704R2

Dear Dr. BHATT,

We’re pleased to inform you that your manuscript has been judged scientifically suitable for publication and will be formally accepted for publication once it meets all outstanding technical requirements.

Kind regards,

Rajesh Raushan, PhD

Academic Editor

PLOS ONE
---

## [Editor Report · Acceptance letter]

11 May 2024

PONE-D-22-32704R2 

PLOS ONE

Dear Dr. Bhatt, 

I'm pleased to inform you that your manuscript has been deemed suitable for publication in PLOS ONE. Congratulations! Your manuscript is now being handed over to our production team.

Kind regards, 

on behalf of

Dr. Rajesh Raushan 

Academic Editor

PLOS ONE